# Assessment of Human Health Impacts from Invasive Pufferfish (Attacks, Poisonings and Fatalities) across the Eastern Mediterranean

**DOI:** 10.3390/biology13040208

**Published:** 2024-03-23

**Authors:** Aylin Ulman, Abdel Fattah Nazmi Abd Rabou, Sara Al Mabruk, Michel Bariche, Murat Bilecenoğlu, Nazli Demirel, Bella S. Galil, Mehmet Fatih Hüseyinoğlu, Carlos Jimenez, Louis Hadjioannou, Ali Rıza Kosker, Panagiota Peristeraki, Adib Saad, Ziad Samaha, Maria Th. Stoumboudi, Tarek A. Temraz, Paraskevi K. Karachle

**Affiliations:** 1Mersea Marine Consulting, 48300 Fethiye, Türkiye; merseamed@gmail.com; 2Departments of Biology and Marine Sciences, Islamic University of Gaza, Gaza P.O. Box 108, Palestine; arabou@iugaza.edu.ps; 3Nursing Department, Higher Institute of Science and Technology-Cyrene, Shahat 6036, Libya; sara.almabruk@omu.edu.ly; 4Department of Biology, Faculty of Arts and Sciences, American University of Beirut, Beirut P.O. Box 11-0236, Lebanon; michel.bariche@aub.edu.lb; 5Department of Biology, Faculty of Science, Aydın Adnan Menderes University, 09010 Aydın, Türkiye; mbilecenoglu@gmail.com; 6Institute of Marine Sciences and Management, Istanbul University, Fatih, 34134 Istanbul, Türkiye; ndemirel@istanbul.edu.tr; 7The Steinhardt Museum of Natural History, Israel National Center for Biodiversity Studies, Tel-Aviv University, Tel Aviv 69978, Israel; bgalil@tauex.tau.ac.il; 8Faculty of Maritime Studies, University of Kyrenia, 99320 Girne, Cyprus; fatih.huseyinoglu@gmail.com; 9Enalia Physis Environmental Research Centre, 2101 Nicosia, Cyprus; c.jimenez@enaliaphysis.org.cy; 10Cyprus Marine and Maritime Institute, 6023 Larnaca, Cyprus; louis.hadjioannou@cmmi.blue; 11Department of Seafood Processing Technology, Faculty of Fisheries, Cukurova University, 01330 Adana, Türkiye; alirizakosker@gmail.com; 12Institute of Marine Biological Resources and Inland Waters, Hellenic Centre for Marine Research, P.O. Box 2214, 71300 Heraklion, Greece; notap@hcmr.gr; 13Directorate of Scientific Research and Publishing, Al-Manara University, Lattakia HQ28 RFM, Syria; adib.saad@manara.edu.sy; 14General Fisheries Commission for the Mediterranean (GFCM—FAO), Palazzo Blumenstihl, Via Vittoria Colonna 1, 00193 Rome, Italy; 15Institute of Marine Biological Resources and Inland Waters, Hellenic Centre for Marine Research, 576 Vouliagmenis Ave., 16452 Argyroupoli, Greece; mstoum@hcmr.gr; 16Department of Marine Sciences, Faculty of Science, Suez Canal University, Ismailia 8366004, Egypt; ttemraz@yahoo.com

**Keywords:** invasive alien species (IAS), public health policy, Suez Canal, tetrodotoxin (TTX), Tetraodontidae

## Abstract

**Simple Summary:**

Invasive alien species are a global threat to biodiversity, ecosystem services (e.g., tourism and fisheries) and human health. Pufferfishes are notorious for their toxicity and may also display aggressive behavior, thus posing a threat to human health. In this work, we collected records of attacks, poisonings and fatalities, due to *Lagocephalus sceleratus* and *Torquigener hypselogeneion*, in the Eastern Mediterranean Sea, for the years 2004–2023. These records were retrieved from social media, and the grey and scientific literature, as well as from personal observations. Overall, 198 events impacting human health were documented: 28 records of physical attacks, at least 143 non-lethal poisoning episodes, and 27 human fatalities resulting from consumption. The number of recorded incidents greatly increased after 2019, especially with regards to poisonings, yet whether related to greater media attention, or to increased abundance is unclear. This is a baseline work for future monitoring and raising awareness. Thus, we urge the continuation of national campaigns to caution residents and tourists of these species’ high toxicities and potential aggressiveness.

**Abstract:**

The silver-cheeked toadfish *Lagocephalus sceleratus* (Gmelin 1789), and to a lesser degree the orange spotted toadfish *Torquigener hypselogeneion* (Bleeker, 1852), pose threats to human health from physical attacks and poisonings in the Eastern Mediterranean Sea. This study reviewed human health-related impacts resulting from these pufferfish, compiling and assessing records from online sources, the peer-reviewed literature, medical records, personal interviews, and observations across the Eastern Mediterranean in the years 2004 to 2023. A total of 198 events impacting human health were documented: 28 records of physical attacks, at least 144 non-lethal poisoning episodes, and 27 human fatalities resulting from consumption. The majority of the reported incidences occurred in Syria, Türkiye, and Lebanon. Most physical attacks occurred in summer, while most poisoning events occurred during winter. The number of recorded incidents greatly increased after 2019, especially with regard to poisonings, yet whether this is related to greater media attention, or to increased fish abundance is unclear. This is the first comprehensive study to collate findings on attacks, poisonings and fatalities caused by these pufferfish in the Mediterranean Sea, and may help in improving national health policies. We urge the continuation of national campaigns to caution residents and tourists of these species’ high toxicities and potential aggressiveness.

## 1. Introduction

The Mediterranean Sea, which is highly biodiverse [1], is the world’s most invaded marine region, with over 1000 alien species [2]. Many alien species live in shallow waters, where they are likely to interact with humans, causing negative effects on human health and well-being [3]. These threats can be catastrophic, with IAS considered the second major contributor to native species extinctions, surpassed only by the effects of biological resource exploitation [4].

Six of the thirteen pufferfish species recorded in the Mediterranean Sea entered through the Suez Canal, the remainder being of either East Atlantic, subtropical or circumtropical origins [5]. The silver-cheeked toadfish, *Lagocephalus sceleratus* (Gmelin 1789), was first recorded in the Mediterranean Sea in 2003 [6] and has spread across the basin [5,7]. The presence and expansion of pufferfish species have raised basin-wide concerns, due to the threats they pose to local biodiversity, fishers’ livelihoods, and fisheries, as well as to human health. *Lagocephalus sceleratus* is known to reduce native fish and some invertebrate stocks (e.g., cephalopods [8]), depredate and spoil catches by spearfishers and fishers, destroying both catches and gear [9,10]. Some pufferfish species are highly poisonous. *Lagocephalus sceleratus* is considered to be among the most harmful invasive alien species (IAS) in the Mediterranean [11,12], due to its spread, abundance, and threats posed to the environment, economy and human health. Although its threats are regularly publicized across the region, the number of cases and hence threat level (towards human health) have never been enumerated before. Sümen and Bilecenoğlu [13] report a case of aggressive behavior resulting in the first amputation record from a pufferfish towards a human. The small-sized, orange spotted toadfish *Torquigener hypselogeneion* (Bleeker, 1852), has also spread in the Eastern Mediterranean [14] and hosts the highest toxicity levels along with *L. sceleratus* [15,16,17].

Impacts of invasive species must be better understood to track their threat levels over time and space. This is the first attempt in compiling data of human health interactions (attacks on humans and poisonings) from the silver-cheeked toadfish *Lagocephalus sceleratus* and the orange-spotted pufferfish *Torquigener hypselogeneion* from across the Eastern Mediterranean region, albeit a highly conservative one as most interactions go underreported. First, some background information is provided to explain some special traits pufferfish possess that likely aid in their invasion success. In this study, both *L. sceleratus* and *T. hypselogeneion* (Figure 1) are referred to as pufferfish, except in cases that mention a specific species, when they are referred to by their scientific name. A thorough understanding of these incidents supports the development of this baseline database of events, for assessing spatial and temporal threat variations, and for developing improved preventative measures and outreach for the greater Mediterranean region.

## 2. Background Information

### 2.1. Teeth

The Tetraodontidae family, belonging to the pufferfish lineage, gets its name for its unique dental arrangement (Greek tetra-odóntas meaning four-toothed). Their dentition is notably advanced, characterized by the ability of the teeth to continually regenerate through stem cell growth. Over time, the teeth continuously develop new layers, enhancing their strength [18]. The two upper and two lower teeth are fused to form a robust beak (Figure 1) that closely resembles that of a parrot, resulting in a sharp cutting edge, unparalleled in any other teleost group [19]. The beak can effortlessly slice through hard objects, such as bones, shells, and even metal. Such formidable dentition has granted tetraodontids a broader ecological niche, permitting them to prey on both pelagic and benthic organisms, including those adapted with armour-like exteriors like crabs, sea urchins, and barnacles [5,20,21]. The silver-cheeked toadfish, with its formidable beak (Figure 2), is known to cause significant financial setbacks for fishers, due to both fishing gear destruction and depredation of prey [10,22].

### 2.2. Poison

Some pufferfish pose a serious threat to human health if eaten, due to the high levels of the neurotoxin tetrodotoxin (TTX), found in their tissues. Of approximately 200 pufferfish found worldwide [23], more than 28% are known to contain unsafe levels of TTX for human consumption [24,25,26,27]. Tetrodotoxin acts as a voltage-gated sodium channel blocker in cells [28] and can be very effective at blocking pain at a certain low (and controlled) dosage, but in more severe cases, may cause complete immobility or even death of the victim. Various factors influence TTX levels found in pufferfish, such as species, location, gender, tissue, and maturity stage [26]. There are two primary hypotheses regarding the production of TTX: either it accumulates through the food chain, or it is produced internally by endosymbiotic or parasitic bacteria within the fish [29]. Research indicates that when pufferfishes are fed a TTX-free diet, their toxicity decreases [29], indicating bioaccumulation from the food chain leads to TTX accumulation, but the bacteria hypothesis is not yet ruled out.

#### 2.2.1. TTX Limits

The lethal dosage of pufferfish is categorized based on TTX amounts, with the safety threshold set by the Japanese authorities, at 10 MU/g TTX (equivalent to 2 mg TTX/kg) [30]. It must be emphasized that controversy surrounds acceptable TTX levels in seafood, particularly in the European Union [31]. The EU’s lower limits contrast those of Japan (2.2 µg TTX eq/g) regarding shellfish and gastropods. The European Food Safety Authority (EFSA) Panel on Contaminants in the Food Chain (CONTAM) has determined that a concentration of 44 µg TTX/kg shellfish meat is not expected to cause harmful effects in humans [32]. This threshold value is noticeably more cautious than the pertinent Japanese safety limit for the evaluation of edible pufferfish. A higher concentration of 560 μg TTX/kg of shellfish meat would be expected not to cause adverse effects in humans, according to more recent toxicological data obtained by feeding TTX to mice rather than gavage while using the same calculation methodology as the EFSA Panel. Additionally, the EFSA opinion has advised that due to their similar toxic effects and modes of action, saxitoxins (STXs) and TTXs be merged into a single health-based advice value [32].

Among the 13 pufferfish species in the Mediterranean, *L. sceleratus*, *Lagocephalus suezensis* Clark & Gohar, 1953, and *T. hypsolegeneion* (formerly referred to as *T. flavimaculosus* Hardy & Randall, 1983; [33]) are known to contain TTX above the toxic limit for humans set by Japan, (Table 1), which can be fatal if consumed [16,26,34].

#### 2.2.2. Tetrodotoxin Poisoning Symptoms

Poisoning symptoms usually appear 10–45 min after TTX consumption, However, delays of up to 6 h have also been reported [24,27,35,36]. There are four stages that these symptoms undergo [24]: (1) initial symptoms related to the nervous system and digestive system; (2) limb numbness and paralysis, including pupillary abnormalities and reflex abnormalities; (3) abnormalities related to the nervous system, cardiovascular system, and respiratory system; and (4) severe cognitive and neural deficits that may result in unconsciousness. Poisoning usually results in death within six to twenty-four hours [35]. Patients typically recover without any lasting sequelae if they do not pass away from respiratory failure within 24 h [24,35].

Due to the lack of a known antidote, antitoxin, or treatment procedure for TTX poisoning, cases of poisoning can often lead to death [24]. In cases of poisoning from TTX consumption, the only treatment available to the patient is supportive care [28,35,37,38,39]. Early diagnosis and supportive treatment of TTX poisoning can lead to favorable outcomes [40]. Therefore, it is very important for healthcare personnel to have sufficient knowledge about the clinical findings and complications of pufferfish (TTX) poisoning, so that the health professional can knowingly administer the supportive care.

#### 2.2.3. Pufferfish Toxicity in the Mediterranean

The first pufferfish species recorded in the Mediterranean Sea was *Lagocephalus guentheri* from the Dodecanese Islands in the late 1920’s [41], with its subsequent records not reported until the 1950s from Türkiye, Greece and Israel [42,43,44]. *Lagocephalus guentheri* Miranda Ribeiro, in 1915, was formerly misidentified as *Lagocephalus spadiceus* [45]. Now a common Eastern Mediterranean fish species, *L. guentheri* is non-toxic [34] and has been consumed in the region without any ill effects. *Torquigener hypselogeneion* was first recorded in the Mediterranean in 1987 in Israel but was not found to have a self-sustaining population until 2002 in southwestern Türkiye [46]. These more recent pufferfish invasions in the Mediterranean have led to a combination of notable ecological, economic, and health consequences [7].

Most pufferfish research in the Mediterranean at the onset of these new invasions in the early 2000s pertained to their expanding distribution [6,46,47], while research on their ecological and biological aspects, and investigations into their toxin levels only began shortly after in the late 2000s [8,15,48]. Katikou et al. [15] pioneered the first TTX study in the Mediterranean on *L. sceleratus* tissues (Table 1), and the onset of poisoning occurrences in the Mediterranean coincides with the commencement of toxicological studies. Various methodologies analyzing TTX levels demonstrate *L. sceleratus* toxicity. Further research examined TTX levels in other species, such as *Sphoeroides pachygaster* (Müller & Troschel, 1848), *Lagocephalus lagocephalus* (Linnaeus, 1758), *T. hypselogeneion*, *L. suezensis*, and *Lagocephalus spadiceus* (Richardson, 1845)/*L. guentheri* ([34,49,50], Ulman, unpublished data). These studies demonstrated that *L. sceleratus*, *L. suezensis* and *T. hypselogeneion* are not only highly toxic but can also be lethal to humans (Table 1). In contrast, for *S. pachygaster* and *L. lagocephalus*, instrumental analysis did not detect any TTX ([49,50], Ulman, unpublished data). Most of the pufferfish poisonings and fatalities in the Mediterranean are attributed to *L. sceleratus*, which grows the largest (up to 10 kg) of all the Lessepsian pufferfish species [51].

The toxicity levels of invasive pufferfish vary greatly and are specific to each individual fish specimen. Toxicity may be influenced by underlying conditions in the environment, local bacteria, prey availability and type, season, fish sex and size, and tissue [26]. There are studies reporting that TTX is a reproductive adaptation for pufferfish and shows maximum toxicity in spring and summer, which are the breeding seasons of pufferfish [24,52]. However, Yu and Yu [53] reported that *Takifugu niphobles* and *Takifugu alboplumbeus* were not toxic during the spawning period of pufferfish. The spawning season of *L. sceleratus* in the Mediterranean covers a wide period and can last from April to September [5,54,55]. The findings of studies on the toxin level of *L. sceleratus* in the Mediterranean Sea also show that *L. sceleratus* living in the Mediterranean Sea contain higher TTX in the late autumn months after the spawning period [15,31,34,56,57]. This is a phenomenon that may be specific to the Mediterranean ecosystem [31]. The dangerous TTX is not destroyed by heating the fish meat or organs, and culinary chefs cannot simply remove TTX while preparing the fish. The toxin has been found in every tissue, including the meat. These facts should be highlighted in awareness campaigns, especially considering the confusion that arises from the Japanese tradition of preparing *fugu* from pufferfish.

Designing appropriate management measures for invasive species is challenging, especially in heavily invaded regions, resulting in abundance and impact fluxes. But in order to prescribe the best advice, their threat levels over space and time must be better understood. For some IAS, increasing public awareness and exploring possibilities for their commercial usage can initiate funding for targeted control efforts, especially if these species pose significant threats to marine ecosystems, the economy, or human health [58].

This study is the first to quantify the threat to human health by two invasive pufferfish species in the Mediterranean Sea, providing a baseline analysis of the effects on human health. With the addition of fresh information, this baseline study will be enhanced in order to gain a deeper understanding of behavioral changes in pufferfish and humans.

**Table 1 biology-13-00208-t001:** TTX-level reports of different pufferfish species from the Mediterranean (µg/g).

Species	NI	Analysis Method	Muscle	Gonads	Liver	Intestine	Skin	Reference	Region
*L. sceleratus*	43	ELISA	<1.10–10.1	17.05–239	16.12–88	<1.10–122.25	<1.10–33.75	[15]	Aegean S.
*L. sceleratus*	43	LC-ESI-CID-MS/MS	<0.32–3.47	0.47–46.30	<0.32–44.15	<0.32–37.60	<0.32–1.40	[59]	Aegean S.
*L. sceleratus*	2	ELISA	0.14–0.68	0.68	0.28–5.88	0.51–23.03	0.42–1.80	[60]	Aegean S.
*L. sceleratus*	2	SPR	0.28–1.51	16.74	0.43–31.28	0.43–34.77	0.54–5.32	[60]	Aegean S.
*L. sceleratus*	2	LC-MS/MS	Nd	4.81	8.07	20.6	1.8	[60]	Aegean S.
*L. sceleratus*	2	MBA	2.52	17.05	16.12	56.78	2.42	[60]	Aegean S.
*L. sceleratus*	37	LC-MS/MS	0.2–21.7	1.2–85.2	1.1–239.8	NA	0.3–27.5	[61]	Aegean S.
*L. sceleratus*	16	MBA	<1.1	<2.2–32.15	<2.2–9.95	---	---	[62]	NE Mediterranean
*L. sceleratus*	16	LC-MS/MS	ND–2.83	0.43–52.07	ND–46.18	0.07–7.15	0.13–3.43	[62]	NE Mediterranean
*L. sceleratus*	20	Q-TOF LC-MS	0.70–5.12	0.69–35.60	0.89–21.10	0.79–12.5	2.20–11.80	[62]	NE Mediterranean
*L. sceleratus*	83	LC-MS	0.02–20.72	0.30–189.03	0.04–104.41	---	0.12–18.59	[26]	NE Mediterranean
*L. sceleratus*	110	Q-TOF LC-MS	1.3–7.8	1.4–68.2	0.8–34.2	NA	NA	[31]	NE Mediterranean
*L. sceleratus*	20	LC-MS/MS	0.10–3.42	0.17–80.00	0.12–25.4	0.16–48.8	0.10–3.30	[56]	East Mediterranean
*L. sceleratus*	3	LC-MS/MS	0.10–0.59	0.26–252.97	1.37–46.67	NA	0.15–0.70	[17]	East Mediterranean
*L. sceleratus*	16	ELISA	0.21–8.32	0.32–12.87	0.11–13.48	0.29–11.74	0.16–5.57	[55]	North Cyprus
*L. sceleratus*	1	LC-MS/MS	1.01	25.95	3.08	NA	1.65	[49]	West Mediterranean
*L. sceleratus*	1	LC-HRMS	0.98	25.22	5.36	NA	2.08	[49]	West Mediterranean
*L. sceleratus*	1	ELISA	2.53	33.55	28.3	NA	3.5	[49]	West Mediterranean
*L. sceleratus* (Juvenile)	2	EC MB-based immunosensing	1.40–2.88	NA	NA	2.88	2.59–2.78	[63]	North Aegean S.
*L. sceleratus*	2	LC-HRMS	0.48–2.08	NA	NA	0.73	1.19–1.24	[63]	North Aegean S.
*L. sceleratus*	2	ELISA	1.52–2.33	NA	NA	10.83	2.77–3.18	[63]	North Aegean S.
*L. sceleratus*	3	LC-MS/MS	7.64–36.49	2.13–1324.44	38.92–188.24	34.65–210.87	14.25–63.18	[57]	South Mediterranean
*T. hypselogeneion*	20	Q-TOF LC-MS	15.88–86.07	5.03–100.71	7.04–106.80	12.59–86.30	35.19–139.72	[16]	NE Mediterranean
*L. suezensiz*	20	Q-TOF LC-MS	<0.6–1.44	<0.6–2.02	<0.6–1.44	<0.6–3.09	<0.6–1.91	[34]	NE Mediterranean
*L. suezensiz*	3	LC-MS	<0.6–1.44	0.94–42.0	5.45–5.55	<0.6–3.09	<0.6–1.91	Ulman (unpub. data)	NE Mediterranean
*S. pachygaster*	5	LC-MS	ND	ND	ND	ND	ND	[49]	West Mediterranean
		LOD: 0.05 mg/kg							
*S. pachygaster*	2	LC-MS	ND	ND	ND	NA	ND	Ulman (unpub. data)	East Mediterranean
		LOQ: 0.2 µg/g							
*S. pachygaster*	20	HILIC-MS/MS	<LOQ	<LOQ	<LOQ	<LOQ	<LOQ	[50]	West Mediterranean
		LOQ: 10 µg/kg							
*L. lagocephalus*	14	LC-HRMS	ND	ND	ND	ND	ND	[49]	West Mediterranean
*L. spadiceus*	20	Q-TOF LC-MS	ND	ND	ND	ND	ND	[34]	NE Mediterranean
		LOQ: 0.6 µg/g							
*L. guentheri*	3	LC-MS	NA	ND	0.52	NA	NA	Ulman (unpub. data)	East Mediterranean
		LOQ: 0.2 µg/g							

NI, Number of individuals analyzed. ELISA, Enzyme-Linked Immunosorbent Assay; MBA, Mouse Bioassay. SPR, Surface plasmon resonance. LC-MS/MS, Liquid Chromatography with tandem mass spectrometry. LC-MS, Liquid chromatography–mass spectrometry. LC-HRMS, Liquid chromatography–high-resolution mass spectrometry. LC-ESI-CID-MS/MS, liquid chromatography coupled to electrospray ionization mass spectrometry operating in the conventional mode in addition to low-energy collision dissociation tandem mass spectrometry. Q-TOF LC-MS, liquid chromatography quadrupole time-of-flight mass spectrometry. ND, Not detected. NA, Not Analyzed. LOQ, the limit of quantification. LOD, the limit of detection.

## 3. Materials and Methods

To compile records of pufferfish health hazards (i.e., attacks, poisonings, and fatalities) in the Eastern Mediterranean region, marine scientists specializing in invasive species in Greece, Türkiye, Cyprus, Lebanon, Syria, Israel, Palestine, Egypt, Libya and Tunisia undertook extensive research, probing online news outlets, social media platforms, peer-reviewed articles, hospital records (where available) and the grey literature in the native languages for reports of pufferfish attacks, poisonings, and fatalities specific to these countries (Figure 3). Additionally, where available, personal interview data and observations from co-authors were integrated into our dataset. Records include GPS coordinates (where specified), relevant URLs, and an English language summary. Replicate entries sourced were discarded.

Subsequently, the data were transformed into bar chart and pie charts to represent the ratio of physical attacks, poisonings, and fatalities per state/country, season and year. Some additional unique incidents provided firsthand by some co-authors, which did not strictly align with the predetermined incident categories, are explained in the discussion to better understand some first-hand accounts of pufferfish behavior.

## 4. Results

A review of human health incidents linked to pufferfish in the Eastern Mediterranean from 2004 to 2023 is provided for two categories: poisonings and physical attacks (Figure 4). A total of 198 incidents, including 28 physical attacks (bites), at least 144 non-lethal poisoning, and 27 human fatalities resulting from pufferfish consumption. The largest number of reported pufferfish-related incidents occurred in Syria and Türkiye (Figure 4A), while the highest number of fatalities were recorded in Lebanon. Physical attacks mainly occurred in summer, while poisoning mostly took place in winter. The increase in events over time could be related to improved media attention, or increased abundances, which currently remains unclear.

Details of pufferfish attacks are presented in Table 2, while poisonings and fatalities resulting from the consumption of pufferfish are presented in Table 3. Please note that the mortality events were not double recorded here as intoxication events, although all deaths resulted from severe intoxication cases.

### 4.1. Physical Attacks

Reports of bites or aggressive behavior by *L. sceleratus* seem to be a recent phenomenon. Between 2014 and 2023 Türkiye, Greece and Libya had the most pufferfish attacks (Figure 4A,C). A total of 25 records representing 28 incidents were retrieved (Figure 4A, Table 2), with only four being attributed to *T. hypselogeneion*. Most reports concern bites and occurred in summer (Figure 4B). Some injuries were severe which resulted in a total of three amputations on the extremities, two from Türkiye- a child and a German tourist, and one from Greece (Table 2, records 6, 10, 11, 23). One other notable event pertains to an aggressive approach toward children, underscoring the potential risk pufferfish might pose to this age group under certain conditions; in July 2019, two large sized *L. sceleratus* attempted to attack some children swimming in the shallows, resulting in widespread panic as they ran out of the water, and adults stepped in and killed the two large pufferfish using wooden poles (Hadjioannou pers. obs.).

*Lagocephalus scleratus* feed mostly at dusk and dawn, common knowledge to the fishers that target them, and may not have good visibility under these conditions when the light is changing. However, *T. hypselogeneion* generally spends all day feeding. Despite their relatively small size, they can become ferocious if a new food source is in the vicinity, as observed by several authors on multiple occasions (A. Ulman, C. Jimenez, M. Huseyinoglu, pers. comm.). For example, schooling and going into a frenzied state have been observed when there is freshly speared fish, fish being cleaned from a fishing boat, or during fish collection for research purposes. In synthesis, it is evident that pufferfish remain an ecologically and medically significant species, but their purposedly new attacking behavior is a matter of concern. We do not advise that this attacking behavior be classified an imminent health threat just yet, as such accounts are still rare and are not extreme cases.

In general, *L. sceleratus* shies away from humans, as confirmed across the region by spearfishers, divers and scientists. *Torquigener hypselongeion*, on the other hand, with its very “cute” appearance (small and toy-like), should not be attempted to be touched by people while underwater. It can easily get accustomed to swimmers and divers, as presented in this study (Figure 5).

### 4.2. Poisonings and Fatalities

Consumption of *L. sceleratus* by unwitting fishers and consumers caused poisoning, and in some cases fatalities, ever since its population was established locally. The number of poisonings reported here is an underestimation of the actual extent of the problem. A total of 43 pufferfish poisoning events were recorded from June 2004 to December 2023, affecting at least 144 individuals (Table 3). The earliest records were from Lebanon and Egypt. Poisoning occurred in all surveyed countries, emphasizing its widespread consumption. The largest number of poisonings were reported from Syria, with nine distinct incidents affecting 64 individuals (Table 3, Figure 4A). Most incidents occurred during winter months. Between 2006 and 2023, 14 documented incidents resulted in 27 fatalities; Lebanon alone recorded 16 fatalities (Table 3, Figure 4A,C).

## 5. Discussion

This study is a regional investigation to collect information on attacks, poisonings and fatalities caused by *Lagocephalus sceleratus* and *Torquigener hypselogeneion* in the Mediterranean Sea, including personal observations of scientists involved in this study. The latter are showing that similar cases likely occur but have not been reported to the press, as they are not severe cases. The remaining incidents lack clarity concerning the exact species involved. It is paramount to mention that while most records describe bites of varying severities, a subset showcases attempted attacks that did not result in actual bites. Such behavior suggests that, under certain circumstances, pufferfish might exhibit territorial or provoked behavior [125]. In the case of *T. hypselogeneion*, curiosity might also be playing an important role, as there were reports that they follow divers and approach different objects carried by them, making mouth contact (perhaps for testing the objects—Jimenez pers. obs.). Though *L. sceleratus* typically exhibits a tendency to avoid human contact in the aquatic realm, which is echoed by divers who seldom encounter this species, various accounts suggest that heightened aggressive behavior could be localized to regions and seasons of intense reproductive activity, and may be more aggressive during sunrise and sunset when the species normally feeds. Preliminary interviews with victims of *L. sceleratus* aggression, as well as with fishers, hint at the possibility that vibrant colors and shining objects might act as triggers or attractants. In addition, during IAS clean-up dives, *T. hypsolengeion* was observed to bite lionfish guts, eat *Diadema setosum* gonads, show instantaneous cannibalistic behavior on fatally wounded individuals and lightly bite divers fingers.

Poisoning caused by consumption of pufferfish has been reported with respect to other Tetraodontid species worldwide (e.g., [126,127,128,129,130,131,132]). With respect to *L. sceleratus*, documentation on poisoning cases has been rather sparse in its native region [133,134]. Therefore, the rising number of poisoning records following the species’ spread in the Mediterranean Sea, as well as the continuous consumption and resulting poisonings of other Tetraodontids worldwide, is of concern. The species is marketed in Egypt despite regulations prohibiting its sale [135], yet no fatalities have been recorded, possibly as consumers are familiar with the species’ toxicity, it being common along the Egyptian Red Sea coast. Public awareness campaigns in several Mediterranean countries may account for the relative low number of poisonings, yet persistent and recent records of poisoning and fatalities call for expanding efforts throughout the species reach.

While poisoning records have greatly increased since 2019, fatalities have remained very low, which could be indication of the success of awareness campaign being effective for the general public. However, as the number of tourists is very high in this region, outreach campaigns should continue to be broadcasted in various languages, and should expand to airports, beaches and marinas as people living outside of the region will lack awareness of their threats. At the onset of the invasion, outreach campaigns were widely broadcasted in each nation about the high toxicity and threats of *L. sceleratus*, and it is understood most people are aware of the threats by now. For example, in Greece, five crew members of a foreign ship that was docked in Ierapetra (Crete) were poisoned after consuming *L. sceleratus* they had fished. It would also be beneficial to better understand their change in abundance, hence level of threat, along with ongoing studies on ethology, toxicology, and essential habitats as priority-directed study topics where possible.

Lebanon was the most affected country from human fatalities from pufferfish consumption. Interestingly, Egypt has not recorded any fatalities from pufferfish poisonings, especially since pufferfish are known to be consumed there. Patterns in mortality further reveal an absence of records from Cyprus and Greece. In Greece, the first sighting of *L. sceleratus* dates back to July 2005 [48]. Following this discovery, a strong warning campaign was issued nationally by scientists and later by authorities. Such proactive measures may account for the observed low incidence of poisonings and the absence of fatalities in Greece linked to consumption of *L. sceleratus*. Public awareness campaigns and preemptive strategies may partly explain these results, yet we urge that such campaigns should be expanded to the wider public, in various languages, to avoid intoxications and/or deaths in tourists or newer residents.

The development of pufferfish attacks on swimmers is of very high interest to track and understand (See references in Table 2 and references therein), especially as they increase their abundance into the Central Mediterranean basin, which may develop into a much wider threat. Such attacks may install fear in swimmers and induce negative economic effects to tourism if publicized in an alarming way, but such events certainly need tracking to understand behavior development over time. However, the number of poisonings and fatalities (Table 3 and references therein) seem to be related to national education and outreach campaigns that are broadcasted to the public, in their toxicity and not to consume them. The science detailing the toxicities of pufferfish tissues, and potential for death from consumption is strong; however, the transmission of this fact needs to be clearer and more candid in order to overcome the naivety of people who are aware of this yet still decide to consume pufferfish, which is also a commonality in the region, especially in Türkiye, which the recent incident of a family of seven in December 2023 is a prime example of. Also, many people are misled by the Japanese custom on consuming *fugu*, believing that any pufferfish species can be consumed if prepared correctly by a skilled chef. The Japanese list for allowable pufferfish species which can be prepared for *fugu* does not include the most poisonous species, especially *L. sceleratus*, whereas *Torquigener* species are too small to become a carefully prepared meal. In Japan, where *fugu* is a delicacy, the government reports an average of about 50 poisonings a year, with a ‘few’ of those leading to deaths, attributed to untrained people preparing them at home (https://www.mhlw.go.jp/topics/syokuchu/poison/animal_01.html, accessed on 2 December 2023). The final exam for a fugu chef includes them identifying the different species by sight, ensuring only those allowed on the list can be served.

Geographical discrepancies in incident reporting necessitate a deeper investigation into the possible underlying factors and regional variations including human-fish interactions, potential threats, and the effectiveness of awareness campaigns. Given the heterogeneity of our data sources, which encompass the literature, social media, and both personal observations and communications, these results serve as a conservative estimate, as no central or national authority is collecting records yet, but local scientists (especially those involved here) studying invasive species have been collecting such data for years, and for this baseline data, have contributed to its purpose. Understanding that this baseline is an underestimation will be instrumental in formulating effective management strategies and awareness campaigns, aimed at reducing potential pufferfish-related human health threats in the Eastern Mediterranean.

## 6. Conclusions

This first regional collation of records of health threats posed by invasive pufferfish underscores the urgency of public health measures. This also indicates that the human health impacts resulting from invasive species is an emergent field of research, as invasive species research becomes more concerning and prominent, especially in the heavily affected Eastern Mediterranean region. Ongoing behavioral and toxicity studies are needed, as well as the systematic recording and monitoring of the incidents. We hereby urge anyone with additional records of attacks, poisonings or fatalities to send them to the first and last author of this study for incorporation into the database.

Information campaigns should make both residents and visitors more aware of the risk of consuming pufferfish, and warning of possible aggressive behavior, and medical practitioners, especially emergency physicians, trained to recognize the clinical symptoms. In addition, it is imperative to collaborate efforts across spatial, ecological, public health, and sociological domains for invasive species research, with the ultimate aim of reducing their threats towards human health and well-being.

## Figures and Tables

**Figure 1 biology-13-00208-f001:**
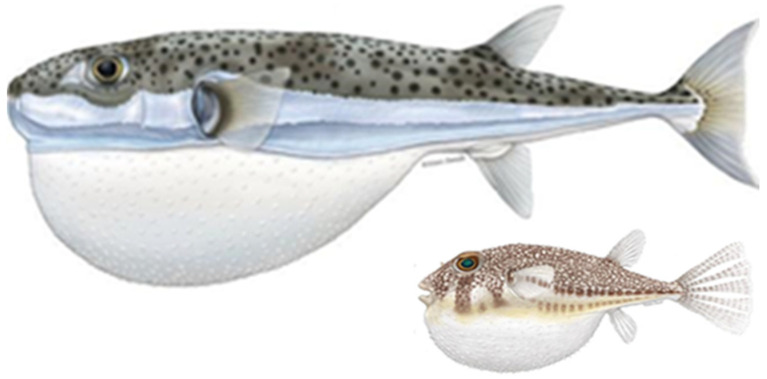
Illustrations of ‘puffed’ *Lagocephalus sceleratus* (**left**) and *Torquigener hypselogeneion* (**right**) not to scale. ©Marc Dando.

**Figure 2 biology-13-00208-f002:**
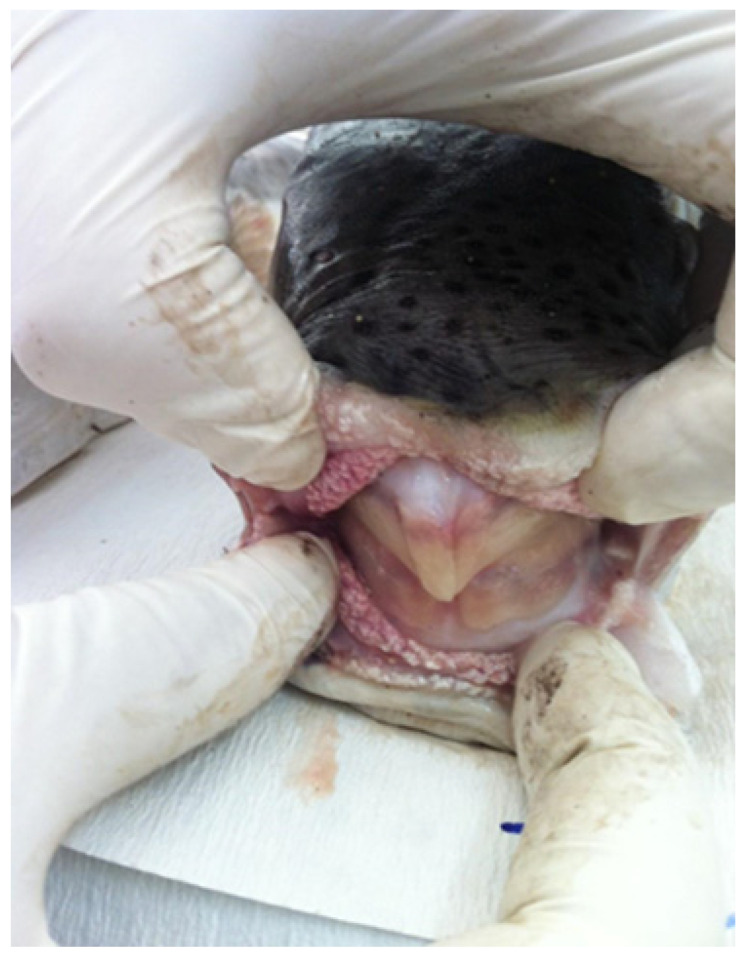
The parrot-like beak of the silver-cheeked toadfish. Photo credit: Ali Riza Kosker.

**Figure 3 biology-13-00208-f003:**
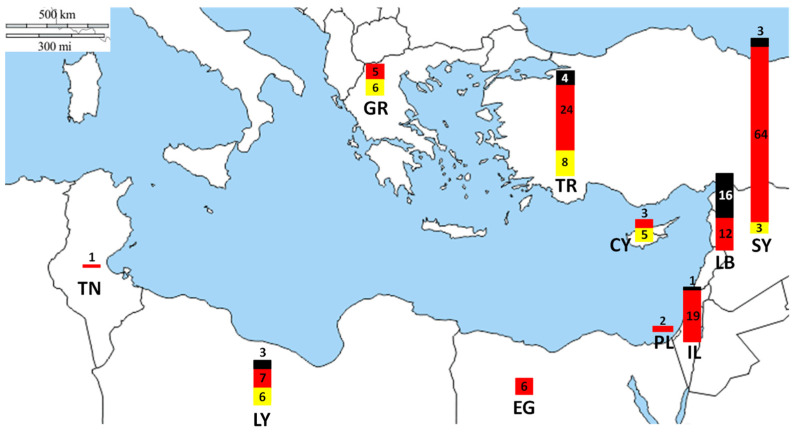
Map of Eastern and Central Mediterranean where records of pufferfish–human interactions were collected from the 10 countries of Greece (GR), Türkiye (TR), Cyprus (CY), Syria (SY), Lebanon (LB), Israel (IL), Palestine (PL), Egypt (EG), Libya (LY), and Tunisia (TN). Yellow bar: attacks; red bars: poisonings; black bars: fatalities.

**Figure 4 biology-13-00208-f004:**
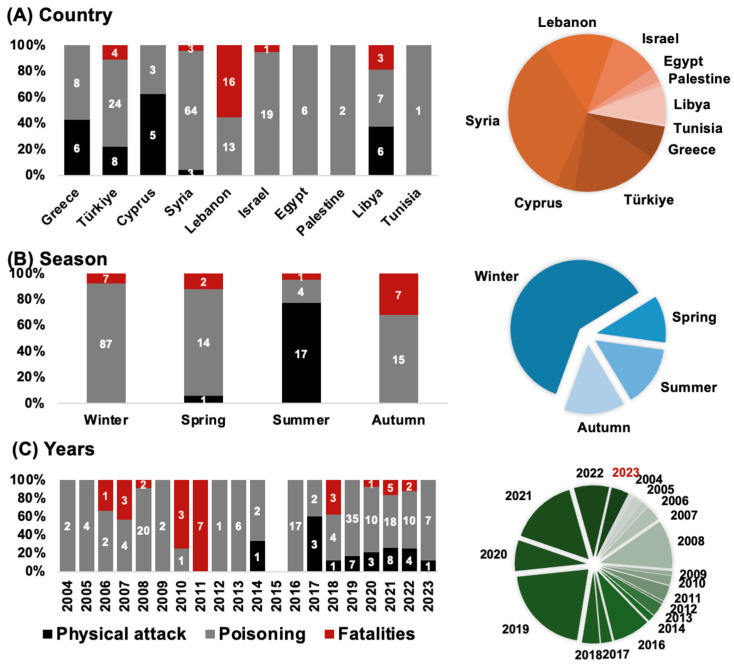
Reported incidents reported for (**A**) Country, Season and (**C**) Years. Bar charts show number and percentages of attacks (black), poisonings (grey) and human fatalities (red); numbers represent incident counts; pie charts show overall incidents (**A**) per country, (**B**) per season, and (**C**) per year. For more details, see Table 2 and Table 3.

**Figure 5 biology-13-00208-f005:**
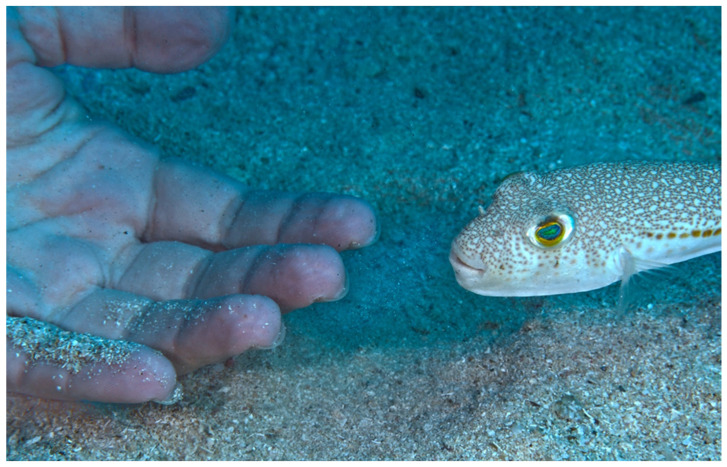
A curious *Torquigener hypsolegenion* from Kaş, Antalya province, Türkiye. Photo credit: Murat Bilecenoğlu.

**Table 2 biology-13-00208-t002:** Records of bites by Mediterranean pufferfish from the literature, social media and from personal observations (pers. obs.) and communications (pers. comm.), NA (no data).

Record #	Country	Incident Date	Incident	Injuries—Type	Species	Reference
1	Greece	2018	1	Bite—Finger	*L. sceleratus*	[64]
2		June 2019	1	Bite—Finger	*L. sceleratus*	[65,66,67,68,69,70,71,72,73,74]
3		August 2019	1	Bite—Heel	*L. sceleratus*	[67,75,76,77,78]
4		June 2021	1	Bite—Large toe	*L. sceleratus*	[79,80]
5		June 2022	1	Bite—Left ring finger	*L. sceleratus*	Peristeraki (interview)
6		August 2022	1	Bite—Right little toe (Amputation)	*L. sceleratus*	Peristeraki (interview)
7		2014	1	Light bites to a diver’s hands	*T. hypselogeneion*	Bilecenoglu (pers. obs.)
8	Türkiye	>2017	1	Bite—Swimmer (back)	*L. sceleratus*	[9]
9		>2017	1	Bite—fisher (finger)	*L. sceleratus*	[9]
10		June 2019	1	Bite—Toe (Amputation)	*L. sceleratus*	[81]
11		August 2019	1	Bite—Child ring finger (Amputation)	*L. sceleratus*	[13,82]
12		June 2020	1	Bite—Fisher	*L. sceleratus*	[83]
13		July 2020	1	Bite	*L. sceleratus*	[84]
14		2021	1	Bite (Inflammation)	*L. sceleratus*	[85]
15	Cyprus	July 2019	1	Bite	*L. sceleratus*	[86]
16		May 2020	NA	Attack—Swimmers	*L. sceleratus*	Hadjioannou (pers. obs.)
17		July 2021	1	Bite—Finger	*T. hypselogeneion*	Huseyinoglu (pers. obs.)
18		2022	NA	Attack—Swimmer	*L. sceleratus*	Hadjioannou (pers. obs.)
19		July 2022	1	Bite—Finger	*T. hypselogeneion*	Huseyinoglu (pers. obs.)
20		August 2022	1	Bite—Finger	*T. hypselogeneion*	Huseyinoglu (pers. obs.)
21		March 2023	1	Bite	*L. sceleratus*	Jimenez (pers. obs.)
22	Libya	2017	1	Bite	*L. sceleratus*	Almabruk (pers. comm)
23		2019	1	Bite—finger (Amputation)	*L. sceleratus*	[87]
24		June 2021	4	Bite—heel	*L. sceleratus*	[88]
25	Syria	2019–2021	3	Bite	*L. sceleratus*	Saad (pers. comm and interview with fishers)

**Table 3 biology-13-00208-t003:** Records of non-fatal poisonings and fatal poisonings (fatalities) by Mediterranean *Lagocephalus sceleratus* from the literature, social media, personal observations (pers. obs.) and communications (pers. comm.).

Country	Incident Date	Non-Fatal Poisonings	Fatal Poisonings	Reference
Greece	December 2013	5		[89]
Türkiye	2008	1		Bilecenoğlu (pers. comm.)
	August 2016	2		[90]
	November 2018	1		[91]
	2019	1		[92]
	2019	2		Bilecenoğlu (pers. comm.)
	May 2020	3		[93]
	July 2020		1	[94]
	January 2021	4	1	[95]
	January 2021	2	1	[96]
	February 2021	1	1	[96]
	December 2023	7		[97]
Cyprus	April 2016	2		[98]
	2017	1		Jımenez & Papagiordou (pers. comm.)
Syria	2005	1		[99]
	September 2007	4		Saad (pers. comm.)
	2010	1		Saad (pers. comm.)
	2016	13		[100]
	2017	1		[81]
	December 2019	30		[101]
	January 2021	7		[102]
	January 2021	4		[103]
	January 2021		2	[102,104]
	April 2022	3		[105]
	April 2022		1	[105]
Lebanon	June 2004	1		[106]
	2005	1		Bariche (pers. obs.)
	September 2005	2		Bariche (pers. obs.)
	2006	1		Bariche (pers. obs.)
	2006–2007		3	[107]
	May 2006	1		Bariche (pers. obs.)
	December 2006		1	[108]
	January 2008	1		[38]
	January 2008	1		[108]
	January 2008	2		[108]
	January 2008	1		[108]
	January 2008		1	[108]
	November 2008	1		[109]
	2010		3	[110]
	October 2011		7	[111]
	February 2022	1		[112]
	April 2022		1	[113,114,115]
Israel	September 2008		1	[116]
	December 2008	13		[117]
	September 2009	2		[118]
	2012	1		[39]
	October 2014	2		[119]
	June 2022	1		[120]
Egypt	December 2004	1		[121]
	May 2022	5		[122]
Palestine	2019	2		Abd Rabdou (per. comm.)
Libya	2018		3	[123]
	February 2020	7		Al Mabruk (pers. comm.)
Tunisia	August 2013	1		[124]

## Data Availability

Data used in this work are included herein.

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
