# Peer review of "Assessment of Human Health Impacts from Invasive Pufferfish (Attacks, Poisonings and Fatalities) across the Eastern Mediterranean"

_biology, 2024, doi:10.3390/biology13040208_

Round 1

Reviewer 1 Report

Comments and Suggestions for Authors

I like this manuscript, it is very comprehensive and well written, and provides valuable information on the negative effects on humans of the arrival of new exotic fish species in the eastern Mediterranean. However, the authors do not mention or compare their results with similar research around the world, with the exception of Japan. For example, a quick check gives us similar information:

Simões et al. Poisoning after ingestion of pufferfish in Brazil: report of 11 cases. Journal of Venomous Animals and Toxins including Tropical Diseases 2014, 20: 54. http://www.jvat.org/content/20/1/54

Gouel, P.; Gatti, C.M.i.; deHaro, L.; Liautaud, A.; Langrand, J.;Boucaud-Maitre, D. Tetrodotoxin Poisoning in Mainland France and French Overseas Territories: AReview of Published and Unpublished Cases. Toxins 2022, 14,351. https://doi.org/10.3390/toxins14050351

 A brief commentary on similar results in other parts of the world should be added to contextualise the importance of this phenomenon worldwide.

Specific comments

Lines 144-152: I don’t understand, if  44 g TTX/kg is greater than 2 mg TTX/kg how can you be more cautious? are the units correct?

Lines 239-241. This paragraph is repeated

Lines 300-303: Move to the discussion section

Lines 337-345: It is not form part of Discussion section, delete or move to the introduction as the aim of the manuscript.

Author Response

The authors would like to thank the reviewer for his/her constructive comments, that have all been addressed (see attached file)

Reviewer 2 Report

Comments and Suggestions for Authors

Generally, I think this manuscript was well organised. The aggression and toxicity of pufferfishes were not only expressed, but their impacts on attacks, poisonings and fatalities were also analysed, all of which can actually improve our knowledge of the threat of pufferfish. Despite this reseach should be of great importance to publish in Biology, a very small puzzle that I still want get the reponse from authors. As you said in context, the toxins of TTX should be tightly related to the dietary of pufferfish, the dietary of fish must be different with season changing. According to your datas, the poisonings and fatalities mainly centralized in winter. Do it mean that the more toxic dietary in winter that can accumulate more TTX in pufferfish. Likelywise, there are also spatial difference in pufferfish diets, the poisonings and fatalities disappeared in part countries, which may be also related to the dietary differences of pufferfishes. Given the importance of dietary differences of pufferfisher to their poisonings and fatalities, I strongly hope you can added some relevant parts to explain the variation of poisonings and fatalities.

Author Response

(The authors gave the same response as above.)

Reviewer 3 Report

Comments and Suggestions for Authors

Presented manuscript by Ulman et al. is review focusing on two dangerous fish, the silver-cheeked toadfish Lagocephalus sceleratus and the orange spotted toadfish Torquigener hypselogeneion in the Eastern Mediterranean Sea. Authors reviewed registered pufferfish impacts on human health from different information sources across the Eastern Mediterranean in the years 2004 to 2023. The work in this direction seems very relevant since there is still a debate in the scientific literature about the reasons for the toxicity of these fish, which may vary regionally and seasonally. The literature is extensive and covers the topic completely, the analysis is carried out at a high level. In general, the work is written in good language and the data presented fully solve the problem, however, a revision, mainly language correction, is required before the article is accepted for publication. Some phrases are difficult to read because they are complex and long; they need to be simplified by dividing them into two sentences. See examples below. 

Line 62. The number of recorded incidents greatly increased after 2019, especially with regards to poisonings, yet whether related to greater media attention, or to increased abundance is unclear. Abundance of fish?

Line 71. Reconsider as “The Mediterranean Sea, which is highly biodiverse [1], is the world's most invaded marine region, with over 1000 alien species [2]. Many alien species live in shallow waters, where they are likely to interact with humans, causing negative effects on human health and well-being [3]”.

Lines 83, 86 Lagocephalus sceleratus change as L. sceleratus.

Lines 164-171 Please reconsider as "Poisoning symptoms usually appear 10–45 minutes after TTX consumption, however delays of up to 6 hours have also been reported [24,27,35,36]. There are four stages that these symptoms undergo [24]: (1) initial symptoms related to the nervous system and digestive system; (2) limb numbness and paralysis, including pupillary abnormalities and reflex abnormalities; (3) abnormalities related to the nervous system, cardiovascular system, and respiratory system; and (4) severe cognitive and neural deficits that may result in unconsciousness. Poisoning usually results in death within six to twenty-four hours [35]. Patients typically recover without any lasting sequelae if they do not pass away from respiratory failure within 24 hours [24, 35]."

Lines 208-210 Please reconsider as "The toxicity levels of invasive pufferfish vary greatly and are specific to each individual fish specimen. Toxicity may be influenced by underlying conditions in the environment, local bacteria, prey availability and type, season, fish sex and size, and tissue."

Lines 221-223 Please reconsider as "This study is the first to quantify the threat to human health by two invasive pufferfish species in the Mediterranean Sea, providing a baseline analysis of the effects on human health. With the addition of fresh information, this baseline study will be enhanced in order to gain a deeper understanding of behavioral changes in pufferfish and humans."

Lines 258-259 Please reconsider as "A review of human health incidents linked to pufferfish in the Eastern Mediterranean from 2004 to 2023 is provided for two categories: poisonings and physical attacks (Figure 4)", etc.

This work, after improving the style, can be accepted for publication.

Comments on the Quality of English Language

Moderate editing of English language required.

Author Response

(The authors gave the same response as above.)
